# Laser-equipped gas reaction chamber for probing environmentally sensitive materials at near atomic scale

**Heena Khanchandani**[1], **Ayman A. El-Zoka**[1], **Se-Ho Kim**[1], **Uwe Tezins**[1], **Dirk Vogel**[1], **Andreas Sturm**[1], **Dierk Raabe**[1], **Baptiste Gault**[1,2], **Leigh T. Stephenson**[1] *

**1** Max-Planck-Institut für Eisenforschung GmbH, Max-Planck-Straße 1, Düsseldorf, Germany, **2** Department of Materials, Imperial College, South Kensington, London, United Kingdom

* l.stephenson@mpie.de

**Data Availability Statement:** Atom probe datasets can be freely accessed using the DOI 10.5281/zenodo.5838237. All other relevant data are within the paper and its Supporting information files.

## Abstract

Numerous metallurgical and materials science applications depend on quantitative atomic-scale characterizations of environmentally-sensitive materials and their transient states. Studying the effect upon materials subjected to thermochemical treatments in specific gaseous atmospheres is of central importance for specifically studying a material's resistance to certain oxidative or hydrogen environments. It is also important for investigating catalytic materials, direct reduction of an oxide, particular surface science reactions or nanoparticle fabrication routes. This manuscript realizes such experimental protocols upon a thermochemical reaction chamber called the "Reacthub" and allows for transferring treated materials under cryogenic & ultrahigh vacuum (UHV) workflow conditions for characterisation by either atom probe or scanning $Xe^+$/electron microscopies. Two examples are discussed in the present study. One protocol was in the deuterium gas charging (25 kPa $D_2$ at 200˚C) of a high-manganese twinning-induced-plasticity (TWIP) steel and characterization of the ingress and trapping of hydrogen at various features (grain boundaries in particular) in efforts to relate this to the steel's hydrogen embrittlement susceptibility. Deuterium was successfully detected after gas charging but most contrast originated from the complex ion $FeOD^+$ signal and the feature may be an artefact. The second example considered the direct deuterium reduction (5 kPa $D_2$ at 700˚C) of a single crystal wüstite (FeO) sample, demonstrating that under a standard thermochemical treatment causes rapid reduction upon the nanoscale. In each case, further studies are required for complete confidence about these phenomena, but these experiments successfully demonstrate that how an ex-situ thermochemical treatment can be realised that captures environmentally-sensitive transient states that can be analysed by atomic-scale by atom probe microscope.

## 1. Introduction

Atom probe tomography (APT) is one of the very few and essential 3D characterization techniques which enable a nanoscale chemical analysis [1, 2]. Over the past few decades since its

**Funding:** B.G. acknowledges support from his ERC-CoG-SHINE-771602 grant (European Research Council - https://erc.europa.eu/). The funders had no role in study design, data collection and analysis, decision to publish, or preparation of the manuscript.

**Competing interests:** The authors have declared that no competing interests exist.

introduction, APT techniques have advanced other materials science areas such as nanoelectronics [3], geoscience [4] and soft matter [5]. But its greatest impact has been from metallurgical APT studies which have investigated the chemical partitioning of phase transformations and solute segregation at defect structures [6]. Bulk properties are determined by both structure and chemistry and so such studies are critically important for many products and industrial applications.

Specific thermochemical treatments lead materials through transient nanoscale states. Analysing these states in their pristine condition is difficult and calls for a suite of supporting protocols. Cryogenic vacuum transfer systems for APT have been developed by several groups for *in situ* studies of environmentally-sensitive materials [7–10]. Integrated treatment cells have been developed to provide immediate APT analysis following a controlled exposure to gases at elevated temperatures [11–15]. Such platforms are capable of exposing materials to hydrogen/deuterium environments at high temperature and subsequently are cooled to freeze the charged structure. Metals such as nickel, cobalt and iron-based superalloys have been investigated in this way [16, 17]. But the procedures that have been used to date require some modification to ensure that a transient state occurring at a high temperature can be faithfully captured within an APT analysis.

The instrument discussed here is called the Reacthub Module and we here considered two examples which have motivated the development of corresponding experimental protocols to operate it, in conjunction with quasi-in-situ subsequent atom probe tomography analysis. One such experiment concerns hydrogen embrittlement of metals during service loading conditions. Several mechanisms have been proposed to cause hydrogen embrittlement such as hydrogen-enhanced localized plasticity, hydrogen induced decohesion, hydrogen-enhanced superabundant vacancies and stress-induced hydride formation [18, 19]. In spite of several studies being undertaken on microscopic scale [20, 21], the actual prevalent hydrogen embrittlement mechanism is not yet well understood. Knowing the exact location of hydrogen atoms (grain boundaries, dislocations or stacking faults) aids in understanding the exact underlying mechanisms which lead to catastrophic failure but as hydrogen is so light and mobile at normal temperatures, it is difficult to capture and observe it with various widely-used microscopic techniques. APT techniques can be used to examine the exact hydrogen trapping sites in the metals but there are various issues in both preparation, data acquisition and data analysis [22–28]. In the current study, we will discuss the example of deuterium charging of a high manganese twinning induced plasticity (TWIP) steel.

Another application concerns the carbon-free direct reduction of iron ore by deuterium and requires similar and careful consideration. Conventional steel-making relies upon the widely implemented blast furnace and basic oxygen furnace processes (70% of the global iron production), resulting in 2.1 tonnes $CO_2$ per tonne of steel [29, 30]. Continued societal development and various technological industries heavily depending on steady supplies of raw iron but appeasing adverse environmental effects in turn requires decreasing carbon emissions. Recent work employing a variety of microscopy techniques by Kim *et al.* [31] have shown for the first time the nanostructures of iron ore before and after hydrogen-based reduction. APT investigations of hematite ore at different reduction stages highlighted the role of impurities and oxygen in the reduction process. Hydrogen's dynamic interaction with the oxide surfaces and interior remained unclear and further investigation required the capability to capture the transient states of an incomplete reduction with finely controlled reduction conditions and an immediate quench. Here, we will present our preliminary results on direct reduction of wüstite by deuterium.

Quantification challenges remain which are here addressed in the experimental steps leading up to APT data acquisition. One challenge is in distinguishing the material's process-

related acquired hydrogen from hydrogen contamination in the APT analysis [32]. This can be tackled by charging the specimens with deuterium (D) and otherwise maintaining clean microscopes, clean specimens, clean specimen holders and fastidious experimental protocols. However, the major issue addressed here is the very high mobility of hydrogen during transfer and preparation of samples. Once the specimen is charged, hydrogen may diffuse out of the specimen during the time taken in the specimen transfer to APT. This issue is resolved by quenching the specimen immediately post-charging so as to cryogenically freeze trapped hydrogen within the specimen's structures. The cryogenic-cooled specimen is subsequently transferred to APT in that quenched pristine state for measurement [33]. We here describe in detail the Reacthub Module which combines pyrometer-controlled laser heating with a Stirling-cryocooled stage and an attached gas manifold to supply treatment.

## 2. Materials

### TWIP steel with composition Fe28Mn0.3C

Twinning induced plasticity steels belong to a class of high manganese steels with Mn content higher than 20 wt. % [34]. They have austenite phase with face centered cubic crystal structure. It is known that TWIP steels are prone to hydrogen embrittlement [35]. Hydrogen embrittlement susceptibility of TWIP steels has been well studied and several mechanisms have been proposed such as influence of hydrogen on stacking fault energy, phase stability and diffusivity [20]. A recent study examined the impact of hydrogen on low cycle fatigue behavior in TWIP steel and postulated that segregation of hydrogen at stacking faults and random high angle grain boundaries in the studied TWIP steel led to its embrittlement [36].

The development of the Reacthub instrument here described provides an experimental solution to test this postulate. Specific features such as grain boundary and stacking faults can be charged with deuterium in the Reacthub followed by immediate quench and subsequently transferred to APT where the trapped deuterium at such structures can be measured. A model TWIP steel with chemical composition of Fe28Mn0.3C (wt.%) was used for current study. The studied TWIP steel was strip cast and homogenized at 1150°C for 2 hours. It was 50% cold rolled and recrystallized at 800°C for 20 minutes, followed by water cooling to room temperature. The charging protocol and preliminary obtained results are discussed in the following sections.

### Single crystalline FeO samples

To start with a model specimen, a single crystal wüstite sample oriented towards the [100] direction, (an orientation accuracy of <0.1° to 0.05°) across its thickness is used. The sample is lab grown through the Czochralski (CZ) method, provided by Mateck GmbH. In the CZ method, the oxide is formed by inserting of a small seed crystal into an oxide melt in a crucible, pulling the seed upwards to obtain a single crystal [37]. Thus, eliminating factors such as impurities and porosity which might be found in commercial ore pellets. The reduction protocol and preliminary obtained results are discussed in the following sections.

## 3. Methods

The two APT instruments that were used for this study were Cameca LEAP 5000 XS and a LEAP 5000 XR both of which had a Ferrovac cryopumped loadlock for attaching the suitcase (detailed below) [9]. APT experiments for deuterium charged specimens were conducted in voltage-pulsed mode, with pulse fraction of 15%, pulse frequency of 200 kHz, set point temperature of 70K and 0.5% detection rate. Data reconstruction and analysis was carried out using

AP Suite 6.0. APT experiments for reduced FeO specimens were conducted in laser-pulsed mode, with laser energy of 50 pJ, pulse frequency of 200 kHz, set point temperature of 45K and 1% detection rate. Data reconstruction and analysis was carried out using AP Suite 6.0.

For the TWIP steel sample, FEI Helios NanoLab 600i dual-beam focused ion beam scanning electron microscope (FIB/SEM) was used for preparing APT specimens containing a region-of-interest for deuterium charging experiments via a standard site-specific lift out procedure [38]. For a wüstite sample, APT specimens in the [100] direction were prepared via standard site-specific lift out procedure [38] using a FEI Helios dual beam Xe-plasma FIB/SEM.

Two ultra-high vacuum (UHV) carry transfer suitcases (Ferrovac VSN40S) were employed [9] for the current study. Cryogenic temperatures were maintained inside the suitcases by liquid nitrogen. The suitcases are designed to hold a modified Cameca APT puck and has a 50-cm long wobblestick which ends with a PEEK-insulated puck manipulator. Inside the suitcase, a $10^{-8}$ Pa pressure can be achieved. The suitcases could be mounted onto the experimental platforms through specially designed load locks (Ferrovac VSCT40 fast pump-down docks), pumped via a 80 L/s turbopump (Pfeiffer HiPace 80).

## 4. Reacthub module

The current work presents the development of the "Reacthub Module" which can be used for heat treatment of APT specimens in a controlled atmosphere. As the name suggests, it is a module separate to other devices but connected to them via a vacuum-carry-transfer suitcase, enabling specimen transfer to the APT and other microscopes without contamination. The Reacthub is equipped with a Stirling cryocooled stage, allowing for passive but rapid specimen quenching after a heat treatment. This treatment is essentially provided by an infrared laser with a red pilot laser for targeting.

### a. Design

The Reacthub is shown in Fig 1(a) and 1(b). It is a UHV system which is broadly comprised of a nominally-UHV reaction chamber attached via a load lock to a suitcase carrying the samples [9]. The Ferrovac suitcases can be attached via a cryopump-equipped load lock allowing for rapid and pristine specimen transfer to and from our Thermofisher Xe-plasma FIB and our two Cameca APT instruments described above. The suitcase is sealed with either an annealed Cu gasket or with a Viton gasket on a CF40 flange and with appropriate care is easily attached or removed in under 5 minutes. The suitcase can hold a specimen puck which is insulated from transfer arms with a polyether ether ketone (PEEK) separator. The suitcase wobblestick is also further insulated with PEEK. When the load lock and the suitcase valves are fully open, sample transfer between the reaction chamber and suitcase was accomplished using the suitcase wobblestick. Upon insertion into the Reacthub, the puck is fastened in place upon the cryostage shown by Fig 1(c), and the load lock valve is then closed. The stage component was custom-made to fit the cryogenic head but in all other respects was engineered along standard designs so as to fit Cameca's atom probe specimen pucks. The Stirling motor (Sunpower Cryo-Tel Cryocooler) cryostage with active vibration dampening, seen to the top of Fig 1(b), achieves temperatures as low as 45K. Cryogenic temperatures are higher in the presence of introduced gases.

### b. Gas manifold systems

Fig 2 shows a schematic for the Reacthub's vacuum and gas manifold systems. When sealed and admitting no gas, the Reacthub is a UHV system. The chambers are pumped to UHV by

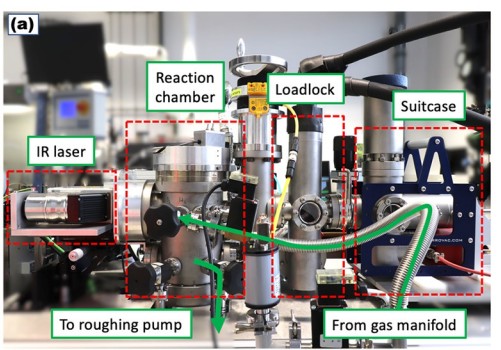
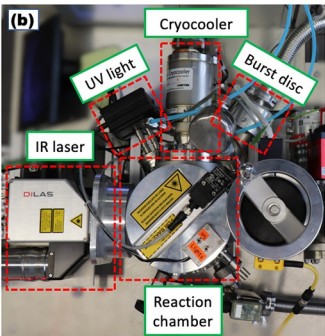
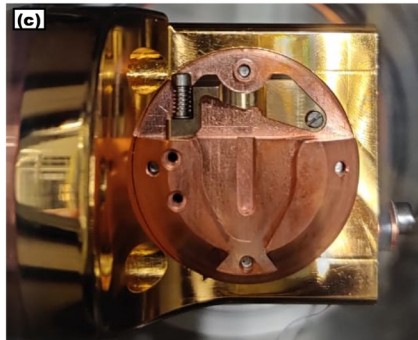

**Fig 1. Photographs of Reacthub module components.** (a) Side profile of Reacthub. (b) Top profile of Reacthub. (c) Top view of the reaction chamber cryostage with the insertion of the atom probe puck from the top.

two turbomolecular pumps (Pfeiffer HiPace-80) which are respectively backed by two multi-stage roots pumps (Pfeiffer ACP-15), each with an $N_2$ gas ballast inlet. The schematic shows the chambers' connection to the pumps, pressure gauges and valves (all of which are manual). An exhaust leads to an emergency burst disc in case of explosive overpressure and care must be taken that the chamber pressure does not exceed 1.5 bar. A sampling line with a heat-jacketed capillary leads to a gas analyser module to measure gas composition. The gas manifold has four lines, three of which are supplied with high purity $H_2$, $O_2$ and $N_2$ and an additional elective line which supplied high purity $D_2$. Gas insertion is through computer-controlled pressure regulators and flushing can be performed by using an additional $N_2$ mass flow controller (Bronkhorst components). It is critically important that strict protocols for chamber and line flushing, gas introduction and evacuation are followed as these are essential to avoid

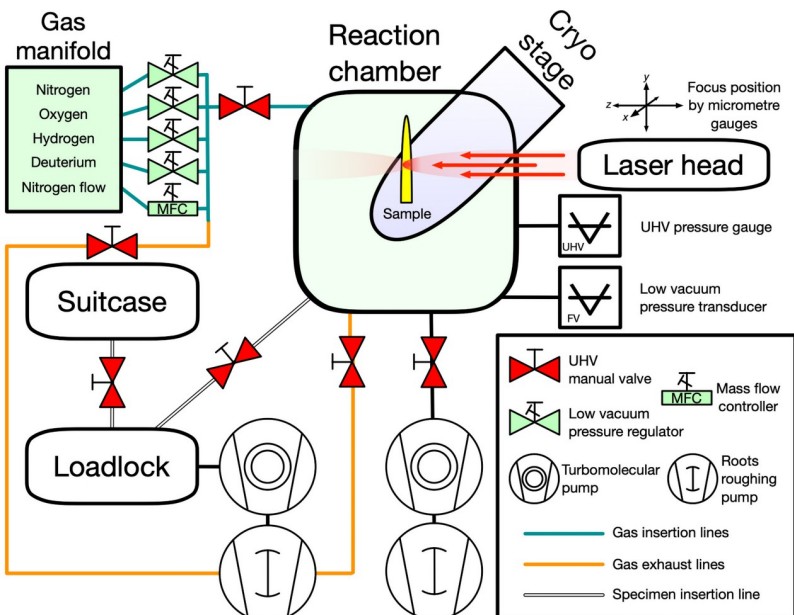

**Fig 2. Reacthub schematic.** The Reacthub consists of an ultrahigh vacuum chamber with a cryostage holding the specimen, an infrared laser which heats the specimen's immediate mounting, and a manifold for controlled filling of particular gases.

unwanted gas contamination and frost formation which will affect the experimental results, even if the compromised sample were to afterwards run at all. We talk about these protocols in a separate section below.

### c. Laser head module

The last essential Reacthub module component is the laser head module which focuses light received from the dual-beam laser unit (DILAS) underneath the workbench via an optical fibre. The laser head module was attached on the insertion axis opposite to the suitcase and directed back towards the mounted specimen. It was mounted on a movable stage controlled by three micrometer screw gauges and, using these gauges, the laser focus can be moved with respect to an inserted specimen. The fixed laser wavelengths are 650 ± 15 nm (pilot) and 808 ± 10 nm (heating), with the latter having a maximum output of 30W. As this is a Class IV laser, interlocks disabling the laser unit are triggered whenever the reaction chamber is opened in any way that would release light. Technically, this makes it a Class I laser product. At the laser focal point, the laser has an approximately 300-μm spot size. Laser focusing with the con-focal pilot laser is best performed only after a steady temperature is achieved to avoid the specimen contracting away.

Together with a camera which aids in positioning, the laser unit has an in-built single-wavelength pyrometer (1800–2100 nm) that has been tuned to the emissivity of grey steel (0.997) meaning that its accuracy may be affected after surface modification by heat and gas treatments. The pyrometer reads from the field-of-view encompassing the grid's laser target and can read in the range 250°C until 1500°C. The upper value was approximately confirmed by melting a steel between 1400–1500°C. The pyrometer provides feedback to laser power controls, allowing for controlled ramping of the specimen temperature. User scripts can provide heat treatments that can be as simple or as complex as desired. The laser's control software permits increasing and decreasing controlled ramps from one temperature to another over a specified period, but this capability was not demonstrated here.

### d. Basic operational protocol

The basic operational protocol follows. The Stirling cooler was activated and the stage was cryogenically cooled. A specimen was transferred onto the cryostage from an attached suitcase, observing UHV protocol. Gas lines were then flushed with $N_2$ and then flushed five additional times with the desired gas, after which the previously ultra-high-vacuum-pumped reaction chamber is populated with the desired gas. We then waited until a stable temperature was reached upon the cryostage as the introduced gas provided an extra load for the cooler and in the subsequent heating from minimum cryogenic temperatures (45K to approximately 140K) the specimen moves with thermal expansion of components. The pilot laser was positioned using the XY-micrometer gauges and focused using the third Z-micrometer. Thereafter, the pilot laser and the chamber lights were switched off. A laser control script can then be run or, alternatively, the laser can either be set to a target temperature or a target laser power expressed as a percentage of the maximum available power. The infrared laser can be immediately switched off following a heat treatment which effects an immediate quench by the pre-cooled stage.

### e. Sample mounts

Specimens could be mounted on a laser-ablated cold-rolled 304 stainless steel (304SS) TEM half-grid (sourced from JPT and CAMECA) shown in Fig 3(a). Each grid was suitable for three lift-out specimens. The grids were 50-μm in thickness and can be held by any suitable

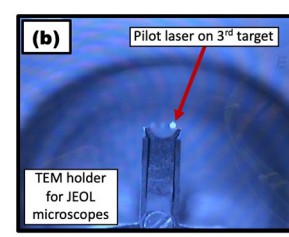
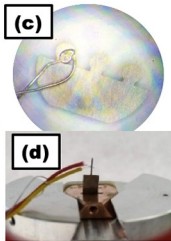

**Fig 3. Sample mountings and thermocouple weldings (a) Schematic of the cold-rolled 304 stainless steel Reacthub grid with 500-μm targets; (b) Laser focus on one grid; (c) Thermocouple spot welded to one of a Reacthub grid's 500-μm targets; (d) A thermocouple attached to the "candle" geometry, here made of a high strength dual phase steel by electrical discharge machining.**

holder [39, 40]. Matching the size of the focused laser spot, each grid had three circular laser targets (either 300 μm or 500 μm in diameter). Above each target sit a prominent mounting wedge. The circular laser targets were isolated from the bulk of the half-grid by a narrow bridge, allowing for specific and efficient heating of only one target to the exclusion of the other targets (see Fig 2(b)), but also wide enough to provide a rapid cooling response when the IR laser power changes or was completely switched off.

The specimen mounting wedges were pre-milled using Xe-plasma FIB so as to provide a clean and discernable area for attaching a sample using a standard lift-out procedure. We developed two mounting formats shown in Fig 4(a). One is the simple conical microtip which allowed for standard lamella lift-outs and normal Pt-welding. The other was a faceted mounting, but required a larger lamella and more Pt-welding to make it secure. The reason for using this latter solution is that the faceted mounting was more robust against the deformation that recrystallization or martensitic transformation can cause in the supporting stainless steel grid.

Fig 4(b) shows another specimen geometry we developed for treating samples which do not necessarily require a site specific lift-out. The samples could be cut from bulk via electrical discharge machining (EDM) into a special geometry here called a "candle". They can be directly placed into the cryo-puck for subsequent controlled treatment in the Reacthub. Because such a material can be different from the 304SS grade used to make the laser-milled half-grids, separate pyrometer calibrations are required before doing the intended experiment.

## 5. Results

### a. Temperature calibrations

Emissivity varies with the steel grade, the surface finish and also upon the sampling wavelength [41]. The pyrometer readings then do not guarantee accurate temperature readings. To confirm pyrometer readings, we spot welded small thermocouples (K-type) to one of the half-grid circular laser targets and one candle sample, as shown by Fig 4(c)/4(d). In this way, we were able to establish curves calibrating the pyrometer and thermocouple readings. These calibrations were obtained under various experimental conditions, such as ultrahigh vacuum, 5-kPa $H_2$ flow or a static 25-kPa $H_2$ atmosphere and with and without stage cooling. Pyrometer and thermocouple readings are abbreviated as "$T_{pyro}$" and "$T_{TC}$" respectively.

Fig 5 shows the calibrations for the 304SS half-grid in 5-kPa flowing hydrogen and 25-kPa static hydrogen atmospheres. An active cryostage was used as many hydrogen experiments demand a rapid quench to cryogenic temperatures to capture fast diffusing hydrogen or a transient transformation state. Fig 5(a) shows three consecutive calibration experiments in a continuous 5-kPa flow. A spline fit made through the data ensemble was not a straight line. One

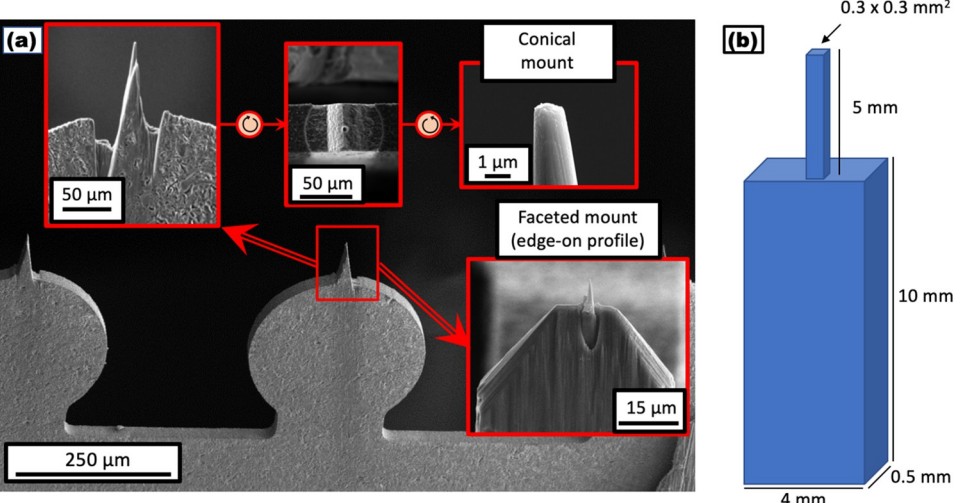

**Fig 4. Pre-milling of sample mountings (a)** SEM images showing the preparatory focused ion beam milling upon a SS304 Reacthub grid (300-μm target version), demonstrating two mounting preparations–a simple conical mounting format and a more elaborate faceted mounting; **(b)** Schematic of "candle" geometry fashioned by electrical discharge machining for bulk specimens.

reason for this could be surface changes in the steel half-grid owing to the first of the three high temperature hydrogen exposures. Thus the data contains two useful calibrations. The initially acquired set can be used for low temperature applications (fit: $T_{pyro} = 1.15 \cdot T_{TC} + 17.46$) and the last set (fit: $T_{pyro} = 1.03 \cdot T_{TC} - 7.42$) can be used for higher temperature applications

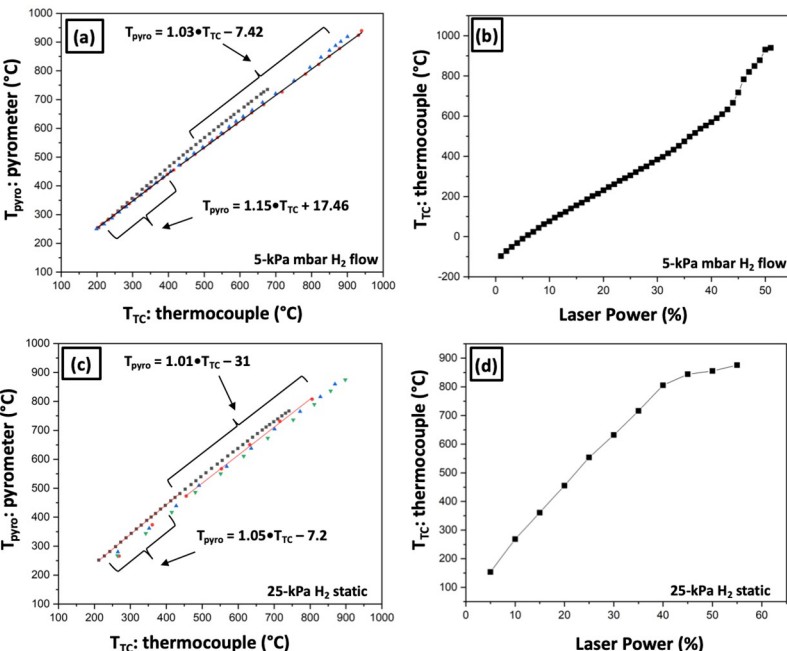

**Fig 5. Pyrometer-thermocouple calibrations.** Calibration curves for the Reacthub 304SS samples grid in two types of hydrogen atmospheres. (a) The pyrometer vs thermocouple calibration curve for three consecutive heating experiments in a continuous 5-kPa $H_2$ flow. (b) The corresponding laser power vs thermocouple reading curve (1st cycle). (c) The pyrometer vs thermocouple readings for three consecutive heating experiments in a static 25-kPa hydrogen. (d) The corresponding laser power vs thermocouple reading curve (1st cycle).

($> 450°C$). Alternatively, Fig 5(b) shows the relationship between the laser power and thermocouple readings which can be used to estimate sample temperature given by corresponding laser power. This is particularly useful for temperatures less than 250°C which is below the pyrometer's minimum measurable temperature. Fig 5(c) shows a similar calibration but for the half-grid in a static 25-kPa hydrogen atmosphere. Standard flushing protocols were employed to ensure the purity of the hydrogen gas in the chamber. Like for the 5-kPa condition, a spline fit through three heating cycles does not resemble a straight line. Applying the same logic as before, two calibrations were derived: one for low temperature applications (fit: $T_{pyro} = 1.05 \bullet T_{TC} — 7.2$) and one for higher temperature applications (fit: $T_{pyro} = 1.01 \bullet T_{TC} − 31; > 450°C$). Fig 5(d) depicts the corresponding relationship between laser power and thermocouple reading.

Additional calibrations are contained in the S1 Text. Vacuum annealing of the 304SS sample grid (Fig A in S1 Text) demonstrated a different response without the presence of gas, demonstrating the need for calibrations for each distinct application, but indicating that different cryostage temperatures do not necessarily need to be accounted for. Fig B in S1 Text further demonstrates that, for another high Mn steel fashioned into the candle geometry shown in Fig 4(b), calibrations made under different conditions were all relatively similar indicating that both cryostage temperatures and hydrogen pressure do not necessarily affect the pyrometer readings. Fig C in S1 Text demonstrates that rapid quenching to subzero temperatures in the first second after laser heating is switched off. The fast quenching at the end of the laser annealing process, is regarded as rapid enough to capture transient states in the hydrogen/deuterium charging experiments that are not thought to decay quickly at the intermediate temperatures until cryogenic temperatures.

## b. Deuterium gas charging of the TWIP steels

Fig 6 depicts the protocol for the deuterium charging workflow that we used for investigating deuterium enrichments at grain boundaries. A gallium FIB milled tip with a grain boundary is transferred to the atom probe through atmosphere. As per previous hydrogen charging experiments [32], the sample was heated in the atom probe loadlock at 150°C for approximately 4 hours in order to desorb hydrogen from the sample and the atom probe puck. The LEAP analysis chamber always contains residual hydrogen desorbing from the steel chamber walls [42], and so we acquired data from the same tip before and after charging to distinguish the hydrogen signal coming from charging and that which comes from analysis chamber. After heating this tip was transferred to the analysis chamber and was run to the 3–4 kV range. Then it was transferred via UHV suitcase to the Reacthub for deuterium charging. After six hours of charging and subsequent quenching, it was transferred to the atom probe via UHV suitcase for further measurement.

This protocol was followed recently for charging a high manganese TWIP steel with deuterium gas. Before deuterium charging, which was regarded as approximately equivalent to hydrogen charging (but maybe just a little bit slower), we followed a standard step-by-step site-specific lift-out. This is shown in Fig D in S1 Text.

Fig 7(a) shows the Transmission Kikuchi Diffraction (TKD) of the tip which was subjected to deuterium gas charging experiment described above. The tip contains a Σ3 twin boundary. The pre-charging APT measurement evaporated approximately 3.5 million ions (reaching a run voltage of 4 kV). In a 25-kPa $D_2$ atmosphere, charging was carried out at a laser power of 8% which corresponded to 200°C as per the calibrations (Fig 5(d)). The mass-to-charge spectra of the same tip before and after deuterium charging is shown by Fig 7(b). The pre-charging mass-to-charge spectrum has a peak only at 1 Da whereas the post-charging mass-to-charge

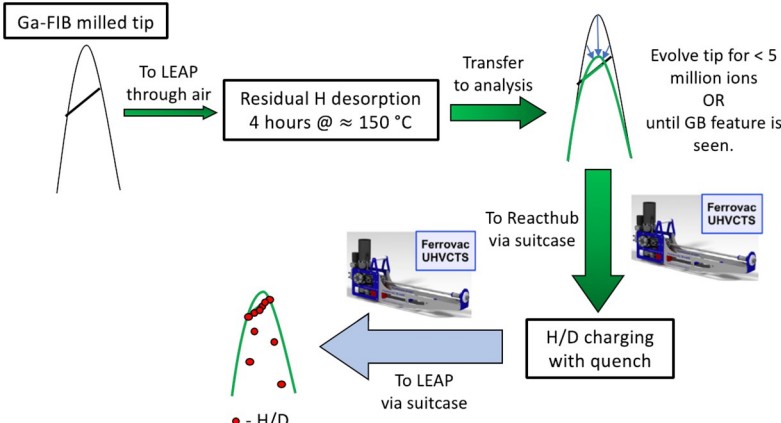

**Fig 6. Reacthub protocol for hydrogen charging.** Workflow schematic for the deuterium gas charging of grain boundary features in high Mn TWIP steels.

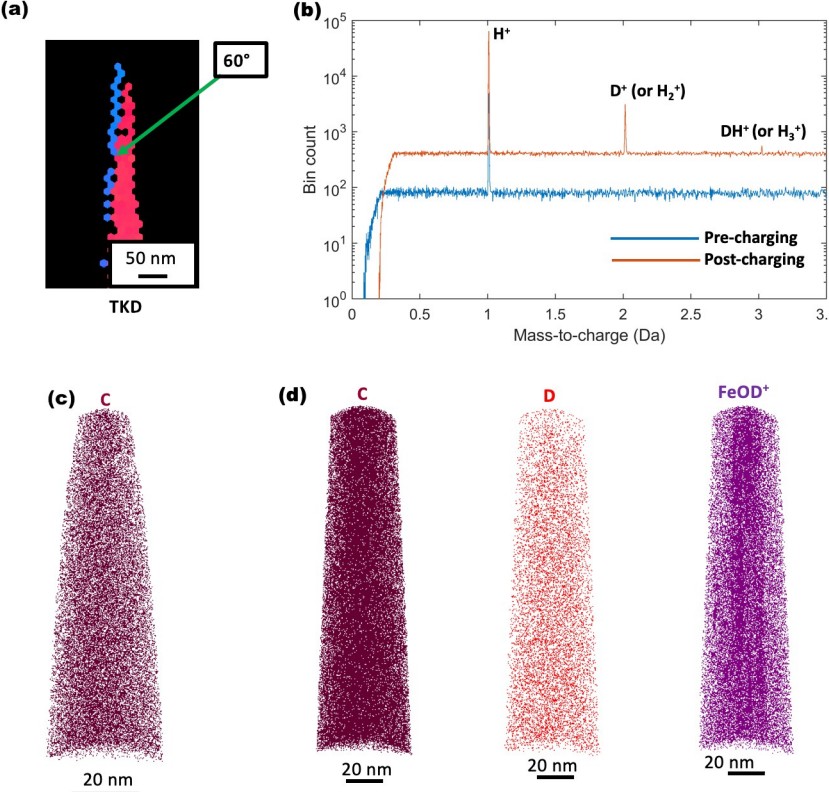

**Fig 7. Characterisation of deuterium-charged high-Mn TWIP steel.** (a) Transmission Kikuchi Diffraction of tip. (b) Pre-charging and post-charging (25-kPa $D_2$ at 200°C) mass-to-charge spectra compared before charging, displaying no signal besides the monoatomic $H^+$. The spectrum after charging displaying peaks at 2 Da and 3 Da taken to be monoatomic $D^+$ and $DH^+$ respectively. (c) Pre-charging distribution of C atoms. (d) Post-charging distributions of C atoms, D atoms and $FeOD^+$ complex ions.

spectrum exhibits peaks at 1, 2 and 3 Da, indicating a change which is here taken to be because of the addition of deuterium. Run under the same electrostatic conditions (voltage mode was used for this reason), the presence of peaks at 2 Da and 3 Da are taken to be $D^+$ and $DH^+$ respectively [43]. The $DH^+$ peak implies perhaps that there could exist a $H_2^+$ (2 Da) and a $D_2^+$ peak (4 Da), but we take the total population of D to be relatively low making $D_2^+$ a rare event. Fig 7(c) depicts a 3D elemental map showing C atoms following the pre-charging reconstruction. The post-charging reconstruction is shown in Fig 7(d), with 3D points maps corresponding to C-containing ions, D-containing ions and the $FeOD^+$ complex ion. C and D atoms are homogeneously distributed, while certain features are enriched in $FeOD^+$ ion. These features could be artefacts associated to damage induced by the energetic electrons used to perform TKD directly on the APT specimen. It has been reported that incoming electrons can knock atoms off their lattice sites and create cascades of structural defects [44, 45]. The precise identification of these features would require further investigation which is beyond the scope of the current study.

### c. Direct reduction of iron oxide (wüstite)

The workflow for the reduction experiments is shown in Fig 8. The plasma FIB was used to make the APT tips that were at least 10 μm in length so as to be readily seen in the positioning optics of the atom probe. The sharpened tips were transferred to the Reacthub and were directly reduced with 5-kPa deuterium at 700˚C for 1 and 2 minutes, respectively. After an immediate quench to cryostage temperatures, the samples were then transferred to the LEAP 5000XS for near-atomic-scale chemical analysis.

Fig 9 shows the reconstructions for these first successful Reacthub reduction experiments. These observations demonstrate that the reduction reaction happens by the diffusion of hydrogen/deuterium into the oxide (and in part also through the first formed Fe layer) instead of being a reaction that occurs exclusively at the specimen surface. The rate of reduction of FeO with hydrogen at 700˚C has been found to be nearly an order of magnitude slower compared to the reduction of hematite and magnetite, due to substantial changes in volume between wüstite and body centered cubic iron and to the possibly slow outward diffusion of oxygen [31]. These specific aspects will be explored in a later study. No evidence here supports such oxygen diffusion out and exploring this will require more experiments at different times of reduction. The 2-min deuterium-reduced sample as shown in Fig 9(c) clearly has a wider portion of Fe than the sample reduced for only 1 min. The deuterium was found mainly at the Fe/ FeO interface. Table 1 also shows the overall compositional analysis of samples at different stages of deuterium-based reduction. After a 2-minute exposure to the reduction atmosphere,

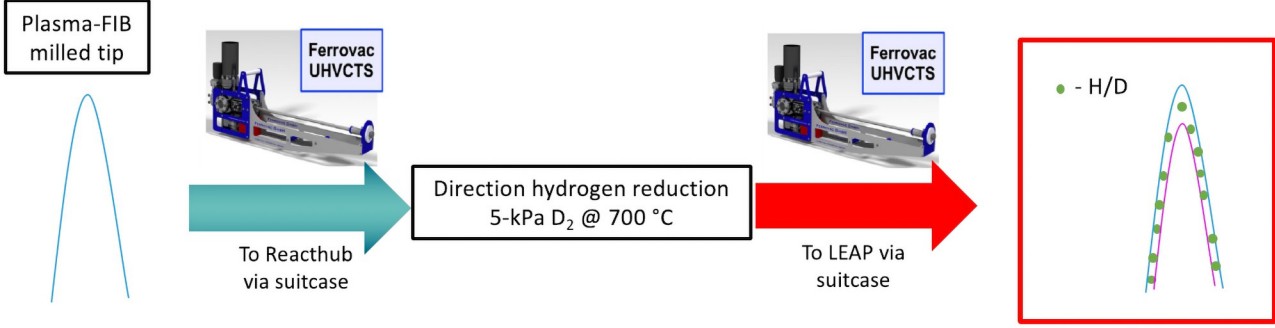

**Fig 8. Reacthub protocol for hydrogen reduction.** Workflow schematic for the direct hydrogen reduction of FeO employing deuterium gas.

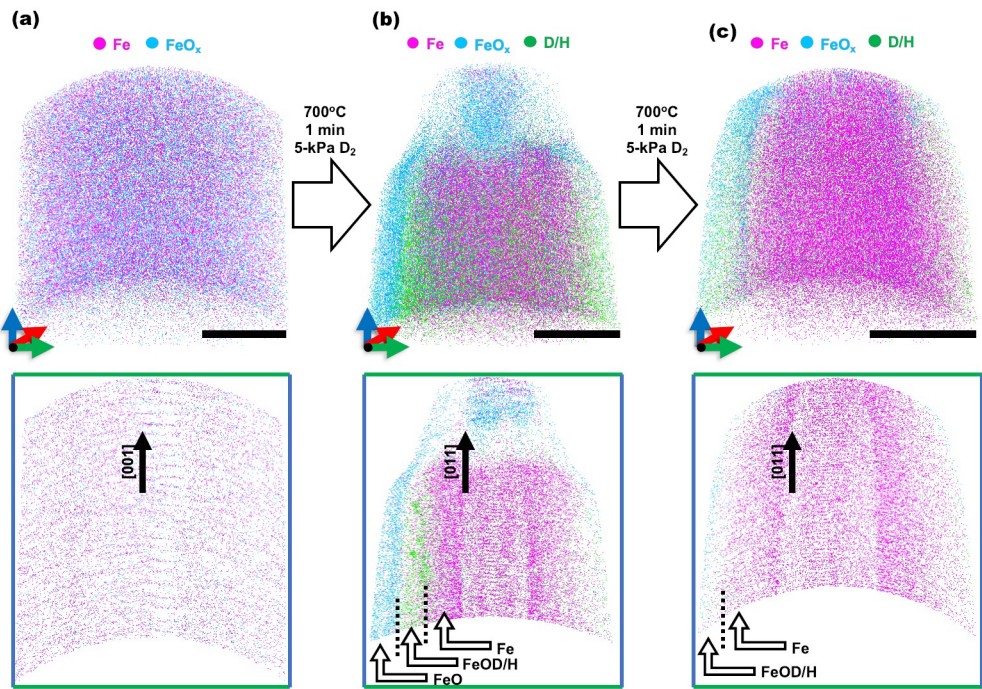

**Fig 9. Atom probe characterization of pristine single crystal wüstite and its reduced states.** Atom maps of FeO, Fe and D/H showing the progress of the deuterium-based reduction and the orientation relationship established by during the reduction (All scale bars are 20 nm).

elemental Fe increases to 77 at. % and the corresponding Fe/O ratio increase demonstrates that the Reacthub can indeed serve to explore subtleties in the reduction process at the atomic scale.

No nano-porosity formation was clear in the reduced atom probe tips. This could be expected as this had been seen in larger samples that were reduced by hydrogen, reported in an earlier study [31]. It is possible that this was due to the different scale of the samples. Porosity in the reduced iron ore pellets was attributed to initial porosity or local gas entrapment whereas, in these nanoscale reduction experiments, the sample had a very high surface-to-volume ratio. The hydrogen and deuterium could interact with the tip surface and diffuse within. This is supported by the planar reduction interfaces seen in Fig 9(b) and 9(c). However, shorter intervals of reduction should clarify further the initiation mechanisms of reduction. As the APT tips in these preliminary results show near-complete reduction after 1 and 2 mins.

The crystallographic planes revealed perpendicular to the poles show that the (100) planes of the parent FeO wüstite phase are parallel to the (110) planes of the subsequently nucleated

**Table 1. Overall compositional analysis (at. %) of APT tips from FeO samples at before and during reduction.**

|  | Fe | O | H/D* | Mn | Si | Al | Mg | Ca | C | Na | Ga |
|---|---|---|---|---|---|---|---|---|---|---|---|
| pristine FeO | 52.8406 | 44.342 | 2.4244 | 0.3055 | 0.0481 | 0.0172 | 0.01396 | 0.0074 | 0.0009 | - | - |
|  | ±0.0197 | ±0.0196 | ±0.0060 | ±0.0022 | ±0.0009 | ±0.0005 | ±0.0005 | ±0.0003 | ±0.0001 |  |  |
| 1 min reduced FeO | 45.7938 | 14.5554 | 38.4769 | 0.1184 | 0.0765 | 0.1269 | 0.0157 | 0.7885 | 0.0059 | 0.0128 | 0.0291 |
|  | ±0.0171 | ±0.0121 | ±0.0167 | ±0.0012 | ±0.0009 | ±0.0012 | ±0.0004 | ±0.0030 | ±0.0003 | ±0.0004 | ±0.0006 |
| 2 min reduced FeO | 77.872 | 5.2342 | 16.7054 | 0.1235 | 0.0254 | 0.0049 | 0.0002 | 0.0058 | 0.0011 | - | 0.0275 |
|  | ±0.0178 | ±0.0096 | ±0.0160 | ±0.0015 | ±0.0007 | ±0.0003 | ±0.0001 | ±0.0003 | ±0.0001 |  | ±0.0007 |

Fe phase. Compositional analysis revealed that the FeO contained at least 0.5 at. % impurities but comparing this to industrial grade FeO made our experiments still count as a model system. The accumulation of the impurities at the reduction interface have been demonstrated previously [31] as one of the aspects that might be relevant when studying the sluggish kinetics of the hydrogen-based direct reduction of iron ores. In our case, preliminary analysis shows some impurities at the interface with the majority forming on the top of the reduced Fe.

## 6. Discussion

The Reacthub Module makes accessible some new atom probe investigations concerning various thermochemical reactions or controlled thermal treatments. Aside from the ability to transport the samples directly to the atom probe following treatment, the Reacthub's cryostage also captures unstable or transient states by enabling a rapid solid-state quench. Careful protocols for the proper flushing of the gas manifold and its lines will likely achieve the purity of the laboratory gas supplies (>99.999% in most cases). In the future, we will be quantitatively measuring this with the attached gas analyzer.

### a. Pyrometer calibrations

Certainty regarding the laser heating was attained by calibrating the pyrometer with a thermocouple. We observed that the laser power changes when changing focus as the pyrometer control feedback attempted to maintain a steady temperature. A crucial first step is always to perform the laser focusing in the same manner for standardized results.

We were able to make estimation of the actual temperature achieved given the pyrometer reading in a specific environmental condition (use of cryostage, pressures of different gases–only for $H_2$ shown here). We only performed the calibrations in vacuum and hydrogen atmospheres and different gases would require a new specific calibration. With exact temperature control the Reacthub does not replicate larger scales as seen in industry and in engineering applications, but instead provides a small-scale foundry with which the fundamental chemical processes can be investigated.

The experiments here used simple thermochemical treatments as model experiments in comparison to what could have been used when aiming at more complex chemical and heating exposure cycles. The calibrations shown in the current study have been performed in vacuum and hydrogen/deuterium atmospheres. They must be performed again for applications involving other atmospheres such as nitrogen and oxygen.

For the deuterium charging of the high manganese TWIP steel, temperatures less than what the pyrometer can measure (< 250˚C) were needed, and laser power vs. thermocouple reading calibration was used. For this, we had to be mindful of the grid's thermochemical processing history which may alter the surface absorbance. For a complex heat treatment, with the applied temperatures varying between the lowest cryostage temperature to >1000˚C and with different gases, a hysteresis investigation would need to be performed. If the experiment is complicated then the calibration must be likewise complicated. An appropriate grid material replacement would be operationally inert.

### b. Grid material

The cold rolled 304 stainless steel was found to be an unideal material for the thermochemical processing considered here. Fig 5(b) and 5(d) showed a change in the laser power-thermocouple coupling at high temperatures, likely made by surface microstructure alterations. Once such change at higher temperatures was seen in Fig 5(b) between 600˚C and 900˚C, where less additional power is required to heat up the target a certain amount. This could be explained by

a higher IR light absorbance of the modified surface and this may also change the surface emissivity used for pyrometer measurements. This can be somewhat accounted for as we did in our work, taking further laser calibrations under the changed surface state.

At the higher 25-kPa hydrogen pressure, the data in Fig 5(d) displayed a different response, slightly decreasing absorbance at 600°C and exhibited a sharp turn to lower efficiency in heating the target above 800°C. This could be because of an increased quenching in a thicker hydrogen environment. But the two conditions corresponding to Fig 5(b) and 5(d) are different in another respect–flowing vs. static atmospheres. The laser power required to reach a particular temperature in the static 25-kPa hydrogen atmosphere (Fig 5(d)) is lower than for a flowing environment at 5 kPa (Fig 5(b)), even when that flow is at 20% the pressure. It was also observed while doing calibrations on candle geometry that a static 100-kPa (1 atmosphere) hydrogen environment consumed less laser power than a 25-kPa hydrogen flowing atmosphere (not shown). This phenomenon may result from an insulation layer that the static atmosphere can provide.

Another issue is that a post milled from the 304SS grid deformed at high temperatures. The faceted mounting was developed to circumvent this problem for the direct hydrogen reduction experiments at 700°C. Using a refractory metal such as tungsten for the half grid may be an option to avoid both deformation and surface modifications but that will require further pyrometer/thermocouple/power calibrations and the pyrometer may even need a factory reset to account for the new material.

### c. Weld stability

A separate question concerns the stability of the Pt-weld at higher temperatures. In the two cases here presented, the weld maintained a sufficient integrity for the successful experiments. How high temperatures may affect the weld and specimen yield is still an open question. Treatment time would also affect the weld. The wüstite reduction at 700°C was only for no more than 2 minutes but this could have induced significant change in the weld structure. Such weld material is electron- and ion-beam decomposed trimethyl(methylcylopentadienyl) platinum$^{(IV)}$ that is decomposed into platinum embedded within an amorphous carbon-rich layer (with a stoichiometry of $Pt(C_8)$ [46]) with methane and hydrogen as the gaseous by-products. Previously, it has been found that a rapid formation of Pt crystallites start nucleating around 580–650°C [47] but that, for a different application, weld stability still held up until 890°C [48]. Also, a study made on amorphous carbon films suggested complete graphitization above 450°C [49] and some loss of mass due to hydrocarbon desorption if the broken material still contains H [50] as perhaps hydrocarbon species do appear in the atom probe spectra of decomposed weld material [10]. This may actually be an issue in a heated hydrogen environment, such as used in both experiments here, which may react with and deplete the amorphous carbon. An alternative would be to use a tungsten precursor $(W(CO)_6)$.

### d. TWIP charging

As per the proposed deuterium gas charging workflow in the current work, the sample was heated in loadlock of the atom probe at 150°C to desorb the hydrogen from sample prior to charging. The deuterium gas charging was conducted at 200°C for 6 hours. A deuterium signal was detected in the post-charging mass-to-charge spectrum. This raises a question on the mechanism of charging. We hypothesize that the ingress of deuterium into the sample takes place during the quenching process. After keeping the sample in deuterium gas atmosphere for 6 hours at 200°C, we immediately quenched the sample on the cryostage by switching off the laser. At this moment, deuterium that was present in the atmosphere of the sample (i.e. in

the reaction chamber) got injected into the sample. In order to confirm this mechanism and optimize the charging process, we are working on further experiments which include charging at lower temperatures or even at room temperature and also for shorter duration.

### e. Wüstite

The results successfully probe into the a couple of the intermediate stages of the direct reduction of FeO by deuterium. Using established Reacthub protocols, we can successfully analyze the gas-solid reactions at the sub-nanometre scale, with the unprecedented capability of allocating deuterium during the progress of the reaction.

Our preliminary data show a core-shell scenario realized by the reaction. The main reactant in this case, i.e. hydrogen, accumulates at the reduction core/shell interface, while the reaction product ($H_2O$) accumulates at the surface. Furthermore, the impact of impurities, even in a lab-made single crystal oxide was found to be present where Na and Mn impurities were allocated at the reduction interface. With only two reduced FeO datasets, no specific conclusions could be made regarding how the reduction kinetic is effected by tip curvature and tip size. This will be addressed in future work by freezing the reduction at smaller times and by varying tip sizes. The initiation of the core-shell structure and its evolution until full reduction will be probed by further experimental and theoretical work. Further Reacthub experiments will yield more about this orientation relationship. No full conclusions can be drawn about the reduction mechanism of FeO, yet the protocols developed in this paper certainly pave the way for more dedicated studies.

## 7. Conclusions

We demonstrated the application of controlled thermochemical treatments for microscopic lamella in a new laser-heated environmental reaction chamber (the Reacthub) with a cryogenic stage for rapid quenching. Pyrometer-to-thermocouple calibrations were performed and it was found that the heating was dependent upon the particular environmental conditions and also upon the laser target's thermochemical history to a certain extent. We showed calibrations for both vacuum and hydrogen/deuterium atmospheres, and found that the 304 stainless steel used for the laser target and sample mounting could be ideally replaced by a more inert material, perhaps a refractory metal. We demonstrated the effective use of those calibrations for two specific examples. Both example studies demand additional experimentation beyond these preliminary runs, but they serve as an adequate demonstration of the Reacthub.

The deuterium charging of a TWIP steel produced a successful experiment that was otherwise inconclusive with respect to deuterium enrichment at grain boundaries. The direct reduction experiments showed an almost complete FeO-to-Fe transformation of the needle-like APT specimens even when only treated for 1 and 2 minutes, but nonetheless revealed the metal-on-metal-oxide reaction interfaces. The quick quench afforded by the Reacthub's cryogenic stage, and the immediate transfer to the atom probe microscope, enabled a nanoscale characterization where deuterium was essentially frozen in place. With conventional specimen treatments, such volatile gases could easily be lost afterward during a slow quench and transfer through air. Together with the rapid response afforded by the laser heating, the Reacthub possesses a unique capability to fabricate as yet unobserved transient nanostructural states and, with aid of the UHV transfer suitcases, transferring the sample to either a plasma focused ion beam microscope or to one of two atom probe microscopes.

## Supporting information

**S1 Fig. Calibration curves for 304SS grid in vacuum (a) Stage at room temperature; and (b) stage at cryogenic temperature.**
(PDF)

**S2 Fig. Calibration for candle geometry.** Calibration curves for a high Mn steel grade for the candle geometry under different conditions.
(PDF)

**S3 Fig. Quenching speed.** Quenching curve depicting the quenching rate of at least 900˚C/s in the first second.
(PDF)

**S4 Fig.** Focused ion beam liftout protocol for atom probe samples (a-c) Site specific liftout procedure for preparing APT tip for D gas charging on the high manganese TWIP steel. (a) A lamella from a sample is cut. (b) The lamella was then lifted out using a micromanipulator and is (c) welded to the mounting grid. (d) The sample is then milled down to appropriate atom probe sample dimensions (<100-nm end radius).
(PDF)

**S1 Text.**
(DOCX)

## Acknowledgments

The authors acknowledge the early contributions of Dr. Thomas F. Kelly (during his time as Division Vice President at Cameca Instruments, Inc.) and Mr. Alexander Rosenthal (Microscopy Solutions).

## Author Contributions

**Conceptualization:** Baptiste Gault, Leigh T. Stephenson.

**Data curation:** Heena Khanchandani, Ayman A. El-Zoka.

**Funding acquisition:** Dierk Raabe, Baptiste Gault.

**Investigation:** Heena Khanchandani, Ayman A. El-Zoka, Se-Ho Kim, Leigh T. Stephenson.

**Methodology:** Heena Khanchandani, Ayman A. El-Zoka, Uwe Tezins, Dirk Vogel, Andreas Sturm, Dierk Raabe, Baptiste Gault, Leigh T. Stephenson.

**Project administration:** Baptiste Gault.

**Resources:** Uwe Tezins, Dirk Vogel, Andreas Sturm.

**Supervision:** Baptiste Gault, Leigh T. Stephenson.

**Visualization:** Heena Khanchandani, Ayman A. El-Zoka, Se-Ho Kim.

**Writing – original draft:** Heena Khanchandani, Ayman A. El-Zoka, Leigh T. Stephenson.

**Writing – review & editing:** Heena Khanchandani, Ayman A. El-Zoka, Se-Ho Kim, Baptiste Gault, Leigh T. Stephenson.

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
