## [Decision Letter · Decision Letter 0]

20 Dec 2021

PONE-D-21-33794Laser-equipped gas reaction chamber for probing environmentally sensitive materials at near atomic scalePLOS ONE

Dear Dr. Authors,

Thank you for submitting your manuscript to PLOS ONE. After careful consideration, we feel that it has merit but does not fully meet PLOS ONE’s publication criteria as it currently stands. Therefore, we invite you to submit a revised version of the manuscript that addresses the points raised during the review process.

We look forward to receiving your revised manuscript.

Kind regards,

Jim P. Zheng

Academic Editor

PLOS ONE

Journal Requirements:

Additional Editor Comments:

Dear Authors:

Manuscript ID PONE-D-21-33794 entitled " Laser-equipped gas reaction chamber for probing environmentally sensitive materials at near atomic scale" which you submitted to the PLOS ONE, has been reviewed. The comments of the reviewer(s) are included at the bottom of this letter.

The reviewer(s) have recommended publication, but also suggest some revisions to your manuscript. Therefore, I invite you to respond to the reviewer(s)' comments and revise your manuscript.

Once again, thank you for submitting your manuscript to the Journal of the PLOS ONE and I look forward to receiving your revision.

Sincerely,

Prof. Jim P. Zheng

Reviewers' comments:

Reviewer's Responses to Questions

**Comments to the Author**

1. Is the manuscript technically sound, and do the data support the conclusions?

Reviewer #1: Yes

Reviewer #2: Yes

2. Has the statistical analysis been performed appropriately and rigorously? 

Reviewer #1: Yes

Reviewer #2: N/A

3. Have the authors made all data underlying the findings in their manuscript fully available?

Reviewer #1: Yes

Reviewer #2: Yes

4. Is the manuscript presented in an intelligible fashion and written in standard English?

Reviewer #1: Yes

Reviewer #2: Yes

5. Review Comments to the Author

Reviewer #1: In this paper, a new thermochemical reaction chamber named the "Reacthub" is designed and established to study the effect of thermochemical treatments on materials in specific gaseous atmospheres, which also allows for transferring treated materials under cryogenic & ultrahigh vacuum (UHV) workflow conditions for characterisation by either atom probe or scanning Xe+/electron microscopes. This work would contribute to the quantitative atomic-scale characterizations of environmentally sensitive materials and their transient states. However, the topic addressed in this manuscript is not within the scope of PLOS ONE. Therefore, I personally do not recommend its publication in this journal of “PLOS ONE”.

Reviewer #2: The paper provides a description of the design and operation a reaction chamber for atom probe tomography. The authors then discuss a couple of case studies to demonstrate the efficacy of this instrumental design. The study is certainly an interesting evolution in the development of in-situ techniques that can be interfaced with atom probe tomography. A couple of items that might be included in the manuscript for those interested in reproducing the studies:

1. What was the effort needed to construct this custom made stage? For instance did the construction of the reaction stage involve much custom made machining ?

2. Does the laser / cryo combination enable one to control the rate of temperature change of the sample?

6. PLOS authors have the option to publish the peer review history of their article (what does this mean?). If published, this will include your full peer review and any attached files.

Reviewer #1: No

Reviewer #2: No

---

## [Author Response · Author response to Decision Letter 0]

24 Dec 2021

Thank you for your careful consideration of our research article entitled “Laser-equipped gas reaction chamber for probing environmentally sensitive materials at near atomic scale”. 

Reviewer 1’s recommendation that the manuscript be refused on the basis that it is beyond PLOSone’s scope is no doubt made under the misapprehension that PLOSone is specifically a journal for the biological or medical sciences. A “big tent” philosophy admitting many diverse fields has allowed this journal to remain competitive. So we thank Reviewer 1 for their expressed concern but find it unjustified. 

Reviewer 2 posed two questions prompting the small changes below.

1. What was the effort needed to construct this custom made stage? For instance did the construction of the reaction stage involve much custom made machining ?

Answered by inserting in Section 4a: The stage component was custom-made to fit the cryogenic head but in all other respects was engineered along standard designs so as to fit Cameca’s atom probe specimen pucks.

2. Does the laser / cryo combination enable one to control the rate of temperature change of the sample?

Answered by inserting in Section 4c: The laser’s control software permits increasing and decreasing controlled ramps from one temperature to another over a specified period, but this capability was not demonstrated here.

We have uploaded the required versions, one showing changes and one without mark-up.

---

## [Editor Report · Decision Letter 1]

28 Dec 2021

Laser-equipped gas reaction chamber for probing environmentally sensitive materials at near atomic scale

PONE-D-21-33794R1

Dear Dr. Stephenson,

We’re pleased to inform you that your manuscript has been judged scientifically suitable for publication and will be formally accepted for publication once it meets all outstanding technical requirements.

Kind regards,

Jim P. Zheng

Academic Editor

PLOS ONE
---

## [Editor Report · Acceptance letter]

17 Jan 2022

PONE-D-21-33794R1 

Laser-equipped gas reaction chamber for probing environmentally sensitive materials at near atomic scale 

Dear Dr. Stephenson:

I'm pleased to inform you that your manuscript has been deemed suitable for publication in PLOS ONE. Congratulations! Your manuscript is now with our production department. 

Kind regards, 

on behalf of

Dr. Jim P. Zheng 

Academic Editor

PLOS ONE